# Evaluating the effectiveness of concurrent opioid agonist treatment and physician-based mental health services for patients with mental disorders in Ontario, Canada

Kristen A. Morin[1], Joseph K. Eibl[2], Joseph M. Caswell[3], Brian Rush[4], Christopher Mushquash[1,5], Nancy E. Lightfoot[6], David C. Marsh[1,4,6,7]*

1 Northern Ontario School of Medicine, Sudbury, Ontario, Canada, 2 Health Sciences North Research Institute, Sudbury, Ontario, Canada, 3 ICES North, Sudbury, Ontario, Canada, 4 Center for Addiction and Mental Health, Toronto, Ontario, Canada, 5 Lakehead University, Thunder Bay, Ontario, Canada, 6 Laurentian University, Sudbury, Ontario, Canada, 7 Canadian Addiction Treatment Centres, Markam, Ontario, Canada

* dmarsh@nosm.ca

**Data Availability Statement:** This study involved third party data from ICES (formally known as the Institute for Clinical and Evaluative Sciences) data

## Abstract

### Objective

The objective of this study was to evaluate the relationship between concurrent physician-based mental health services, all-cause mortality, and acute health service use for individuals enrolled in Opioid Agonist Treatment in Ontario, Canada.

### Methods

A cohort study of patients enrolled in opioid agonist treatment in Ontario was conducted between January 1, 2011, and December 31, 2015, in Ontario with an inverse probability of treatment weights using the propensity score to estimate the effect of physician-based mental health services. Treatment groups were created based on opioid agonist treatment patients' utilization of physician-based mental health services. Propensity score weighted odds ratios were calculated to assess the relationship between the treatment groups and the outcomes of interest. The outcomes included all-cause mortality using data from the Registered Persons Database, Emergency Department visits from the National Ambulatory Care Database, and hospitalizations using data from the Discharge Abstract Database. Encrypted patient identifiers were used to link across databases.

### Results

A total of 48,679 individuals in OAT with mental disorders. Opioid agonist treatment alone was associated with reduced odds of all-cause mortality (odds ratio (OR) 0.4, 95% confidence interval (CI) 0.3–0.4). Patients who received mental health services from a psychiatrist and primary care physician while engaged in OAT, the estimated rate of ED visits per year was higher (OR = 1.3, 95% CI 1.2–1.4) and the rate of hospitalizations (OR = 0.5, 95% CI 0.4–0.6) than in the control group.

analytic services (DAS). ICES is a not-for-profit research organization that gathers population-based health and social data from Ontario's publicly funded health services to generate knowledge. Data availability is subject to the terms of data sharing agreements ICES has signed with David Marsh. Questions on data availability can be directed to das@ices.on.ca.

**Funding:** We received funding from the Northern Ontario Academic Medical Association from the Academic Funding Plan Innovation Fund: project No: A-17-05 to conduct this study. The funders had no role in study design, data collection and analysis, decision to publish, or preparation of the manuscript. This funding contributed to K.A. Morin's postdoctoral fellowship salary.

**Competing interests:** Dr. David Marsh maintains the following roles: Chief Medical Director at CATC (Canadian Addiction Treatment Center), opioid agonist therapy provider. Dr. Marsh has no ownership stake in the CATC as a stipendiary employee. We do not foresee any conflict of interest as data will be made freely available to the public, and neither the CATC, nor the Universities, have the ability to prevent publication and dissemination of knowledge. The other authors have no conflicts declared. This does not alter our adherence to PLOS ONE policies on sharing data and materials.

## Conclusion

Our findings support the view that opioid agonist treatment and concurrent mental health services can improve clinical outcomes for complex patients, and is associated with enhanced use of acute care services.

## Introduction

The opioid crisis continues to have detrimental impacts on communities in Canada despite the increased programming and policies recently put in place to mitigate these effects [1–4]. It was estimated that 15,393 Canadians died from opioid-related causes between January 2016 and December 2019 [5]. In the Province of Ontario, the death rate also continues to rise from 5.2 deaths per 100,000 population in 2016 to 7.9 in 2017 [6]. Additionally, people with Opioid Use Disorder (OUD) are heavy users of emergency departments (ED) and hospitals across the province. For example, in Ontario, opioid-related ED visits increased from an average of 9.42 per 100,000 population in 2003 to 19.55 per 100,000 population in 2015 [7–9].

Importantly, it is estimated that 50–90% of those who develop OUD are also diagnosed with a concurrent mental disorder [10]. The most prevalent mental disorders reported in this population are anxiety and mood disorders, including major depression and bipolar disorder [11, 12]. It is essential to highlight the potential underestimation of the prevalence of post-traumatic stress disorder (PTSD) since it is known to be often misdiagnosed as another anxiety, depression or other related disorders [13–16]. Concurrent mental disorders can be a marker for greater clinical complexity [10, 17, 18]. Bogdanowicz et al. demonstrated that the risk of death is almost four times greater in patients with OUD and concurrent mental disorders [17].

In rural and remote communities in northern Ontario, the opioid crisis is especially critical. Northern communities experience some of the highest mortality rates due to opioids [2] and some of the lowest access to specialized addiction and mental health services [19, 20]. For example, a report by Health Quality Ontario revealed that in 2009, the Toronto Central local health authority had eight times more psychiatrists per 100,000 population than the northern health authorities in 2014 (62.7 in Toronto Central vs. 8.3 and 7.1 per 100,000 in the North East and North West Local Health Integration Networks (LHIN)). Primary care providers and EDs are disproportionately used as a first point of contact for patients in rural areas due to the lack of specialized addiction and mental health services [21].

Fortunately, there are effective medications available through opioid agonist treatment (OAT) programs, including methadone and buprenorphine/naloxone, which are readily available in most regions of Ontario [22]. OAT is commonly made available in specialty fee-for-service clinics. In Ontario, mental health and OAT are funded differently and there is little integration or coordination between both types of services [20, 23, 24]. People with opioid use disorder often present with a complex profile of health and a host of psychosocial issues [25, 26]. Although OAT has been shown to be very effective, its efficacy may be negatively impacted by concurrent issues, including mental disorders [11, 12], psychosocial problems [10–12, 17, 27] and infections such as HIV and viral Hepatitis [28–32]. The prevalence of comorbidities present major public health concerns in terms of the risk of complications from chronic physical health issues as well as poorer health system and treatment outcomes [11, 17, 33, 34]. As the severity of the opioid issue continues to rise, the expansion and integration of OAT with other health services—more specifically, primary care and mental health services—has been recommended as a strategy to mitigate the opioid crisis [35, 36].

There is debate in the literature about whether or not adding mental health services to OAT programs improve outcomes and what is the best type of mental health service to incorporate into OAT programs [27, 37]. Most of the studies in the existing literature only evaluate methadone as a form of OAT, however, the use of buprenorphine/naloxone is on the rise especially in northern Ontario. Many authors have previously focused on psychosocial services, but there are limited studies on physician-based mental health services and it is very common for patients in Ontario, especially in rural areas to access health care from physicians [23]. Moreover, other studies focus on individual clinical settings rather than population wide data. This is important because OAT varies significantly across different settings. Therefore, systematic reviews and meta-analyses are potentially comparing inconsistent control groups. Lastly, the outcomes evaluated in many studies focus on individual clinical outcomes such substance use, retention and treatment complience. But there is limited research evaluating population health such as health system use and all-cause mortality.

Given the high prevalence of diagnosed comorbid mental disorders [10, 17], and the provincial recommendation to integrate OAT with other services, and the shortcomings in existing studies, it is important to understand the intersection of OAT and mental health services in various settings, within the current state of Ontario's health care system. The goal of this study was to compare co-occurring physician-based mental health services and OAT to OAT alone. We hypothesize that patients who received mental health services from physicians while in OAT will have better outcomes than those not receiving mental health services.

## Methods

### Overview

A retrospective cohort study of patients in OAT was conducted between January 1, 2011, and December 31, 2015, in Ontario using propensity score methods. The main cohort included OAT patients who had no methadone or buprenorphine/naloxone prescriptions in the year before the first treatment episode. The first treatment episode was used as the index date for the study, and each patient was followed for one year after their index date. A Patient and Family Advisory Group played a key role in the project by providing insight into factors that directly impact the health of individuals with OUD and mental disorders in Ontario. The advisory group described an accumulation of exposures and experiences such as: stressful life experiences, the need for social support, early life influences, family history of trauma, and experience in accessing health care services, oppression, and shame. The factors described are also supported in the literature [38–42]. The consultation with the advisory group assisted us to choose the exposures and the covariates for this study.

### Study population: First-time OAT patients with concurrent mental disorders

The cohort for this study was created by extracting OAT patients using the Ontario Drug Benefit Plan (ODB) database using drug identification numbers (DIN) and with the Ontario Health Insurance Plan (OHIP) database using physician billing codes including OAT monthly management codes (K682, K683, and K684), visit/consultation codes (A680 and A957) and point of care testing codes (G040, G041, G042 or G043) as well as OHIP diagnosis codes to define mental disorders including neurodevelopmental disorders, schizophrenia spectrum and other psychotic disorders, bipolar and related disorders, anxiety and depressive related disorders, obsessive compulsive and related disorders, trauma and stressor related disorder, feeding and eating-related disorders, gender dysphoria, disruptive impulse control and conduct disorders, and personality

disorders. A published validation study by Davis et al. found that reporting of mental diagnoses have a strong likelihood of being accurately captured in administrative datasets with a positive predictive value of 76% and Kappa showed a moderate agreement (median kappa = 0.45–0.55) [43]. In the OHIP database, substance dependence falls under one category, under one code (304). All patients in the cohort had that code based on their opioid dependence to be included in the study. Therefore, there was no way to detect those who also had a dependence on cocaine, benzodiazepines, or other substances. For this reason, we chose to exclude substance use disorders from our mental disorder diagnosis definition. Mental health diagnostic information is available in S1 Table. The steps to create the cohort are outlined in Fig 1.

## Data sources

The Research Ethics Board approved the study of Laurentian University in Sudbury. The data were obtained by submitting a formal requisition to ICES. ICES is a not-for-profit research

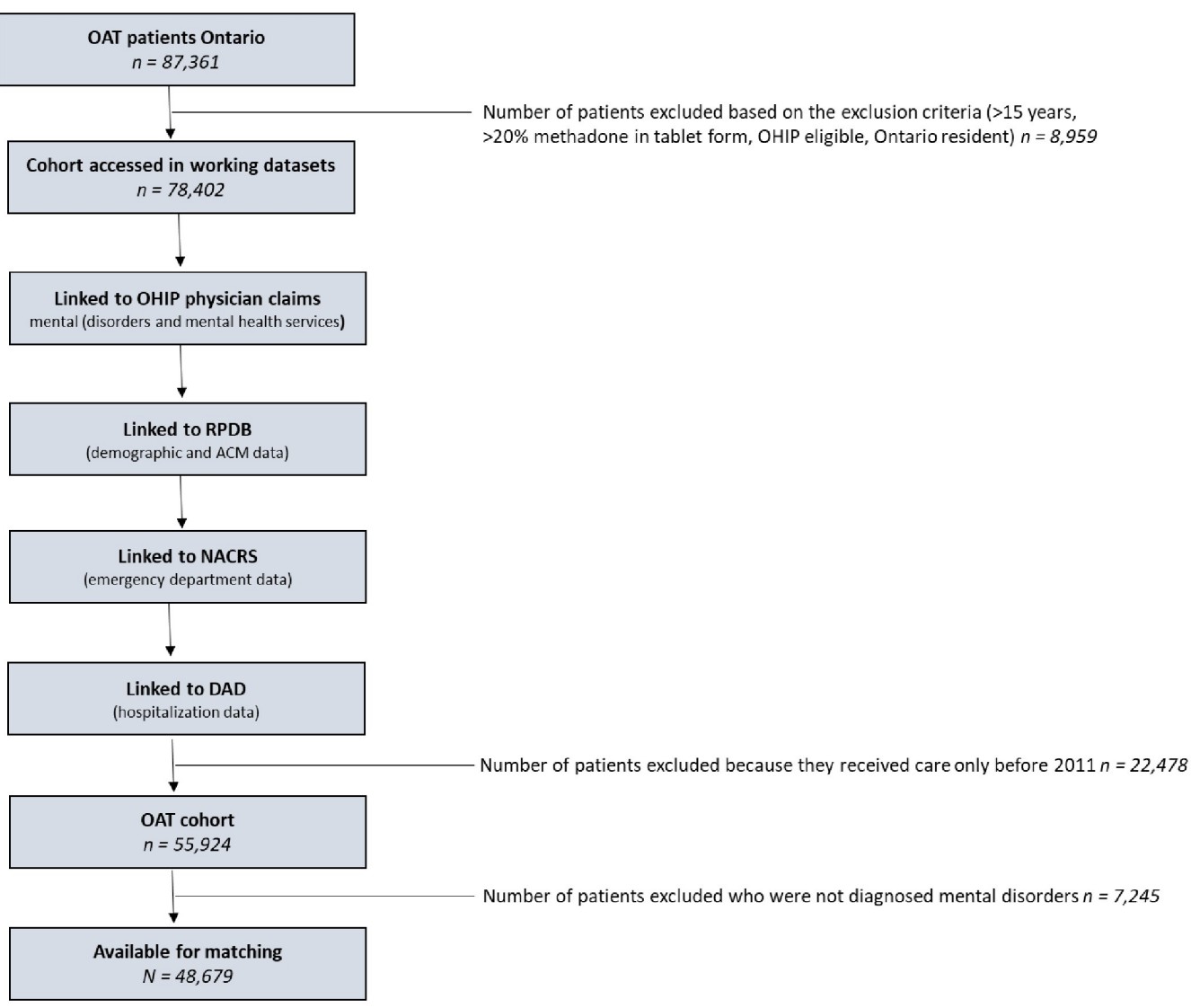

**Fig 1. Flow chart outlining data build including linkages.**

organization that gathers population-based health and social data from Ontario's publicly funded health services to generate knowledge [44]. The data were accessed remotely using a secure server located at Sunnybrook hospital in Toronto, Ontario. ICES provided anonymized individual-level patient data collected from Ontario publicly funded health services.

'The datasets used for analysis included patient demographic information including age, sex, place of residence, neighborhood income quintile and mortality from the Registered Persons Database (RPDB), a record for each inpatient discharge from all publicly funded Ontario hospitals from the Discharge Abstract Database (DAD), a record for each out-patient use of all publicly funded Ontario emergency departments (ED) from the National Ambulatory Care Database (NACRS), a record of each patient publicly funded drug prescription from the Ontario Drug Benefit Plan Database (ODB), a record of each patient encounter with a physician (including any diagnosis given by the physician) from the Ontario Health Insurance Plan Database (OHIP). Patient information was linked across databases using encrypted ten-digit health card numbers. The linking protocol is used routinely for health system research in Ontario [45–49]. Detailed information about databases and study variables are available in Fig 2.

### Exposure

We defined active OAT treatment based on OHIP billing (from the OHIP database). The exact date of service was not provided in the ICES datasets for this study due to privacy reasons. Rather we were able to calculate an approximation of the time of a health service event based on a variable that had information relating to the number of days to service from the index date. As indicated earlier, all variables were linked to an encrypted patient ID. We used the number of days to service date to query whether a patient was enrolled in OAT using 30 day intervals (i.e. if the days to service date was larger or equal to -30 and smaller or equal to 30 at the time that an OAT OHIP billing code appeared then the patient was considered to be

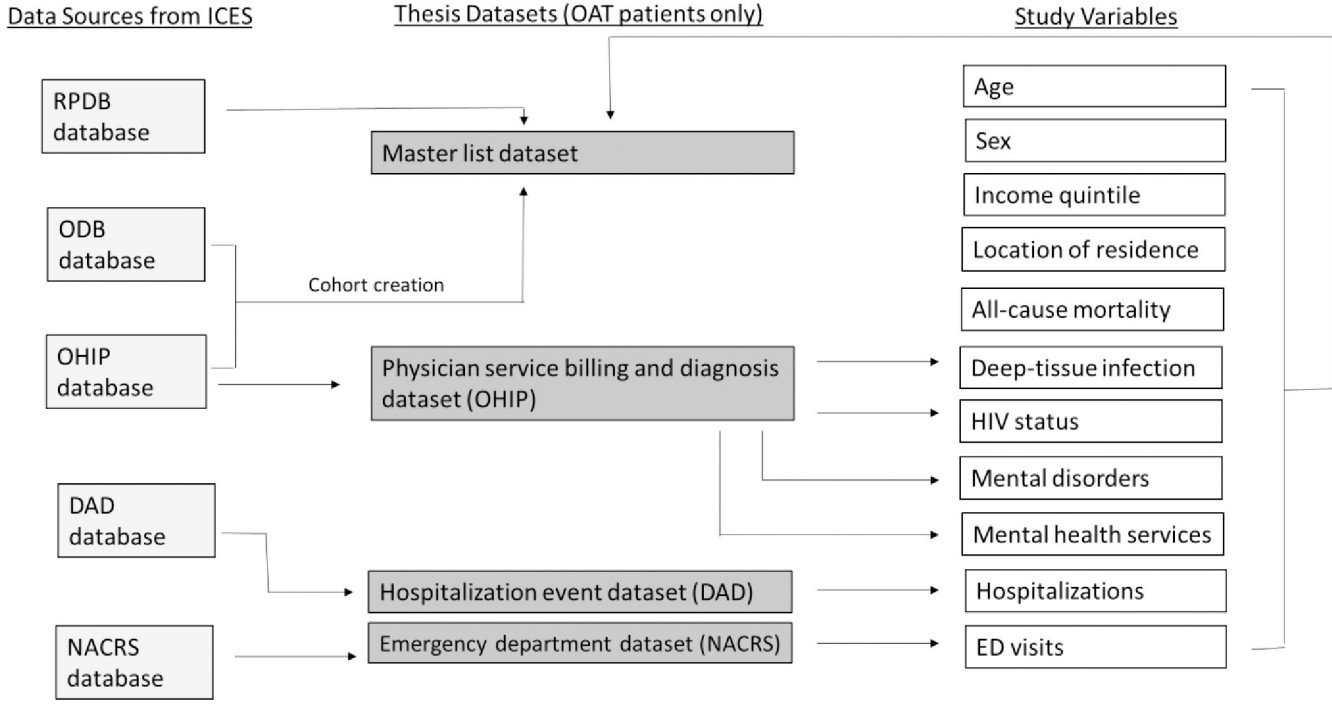

**Fig 2. Databases and study variables.**

"actively enrolled in OAT for at least 30 days, if the days to service date was larger or equal to -30 and smaller or equal to 60 at the time that an OAT OHIP billing code appeared then the patient was considered to be "actively enrolled in OAT for at least 60 days, etc.). The query was repeated in 30 day intervals up to 360 days (one year observation window). We then created a variable indicating whether a patient was on OAT (meaning that there was less than 30 days between the previous OAT billing code) or had received but was not actively on OAT (meaning that there were more than 30 days between the previous OAT billing code) within 360 days of their first OAT visit.

The following mental health service group was created for analysis: Treatment group 0 ($Tx_0$) was the control group for this study and included patients who were not actively engaged in OAT. For instance, patients in this group were not actively seeing a physician and receiving methadone or buprenorphine/naloxone for over 30 days. Treatment group 1 ($Tx_1$) included patients who were actively engaged in OAT but had not received any additional mental health services from a physician. Treatment group 2 ($Tx_2$) included patients who were actively engaged in OAT and had received mental health services from a psychiatrist during their active OAT episode of care. Treatment group 3 ($Tx_3$) included patients who were actively engaged in OAT and had received mental health services from a primary care physician during their active OAT episode of care. Treatment group 4 ($Tx_4$) included patients who were actively engaged in OAT and had received mental health services from both a psychiatrist and a family physician during their active OAT episode of care (Fig 3).

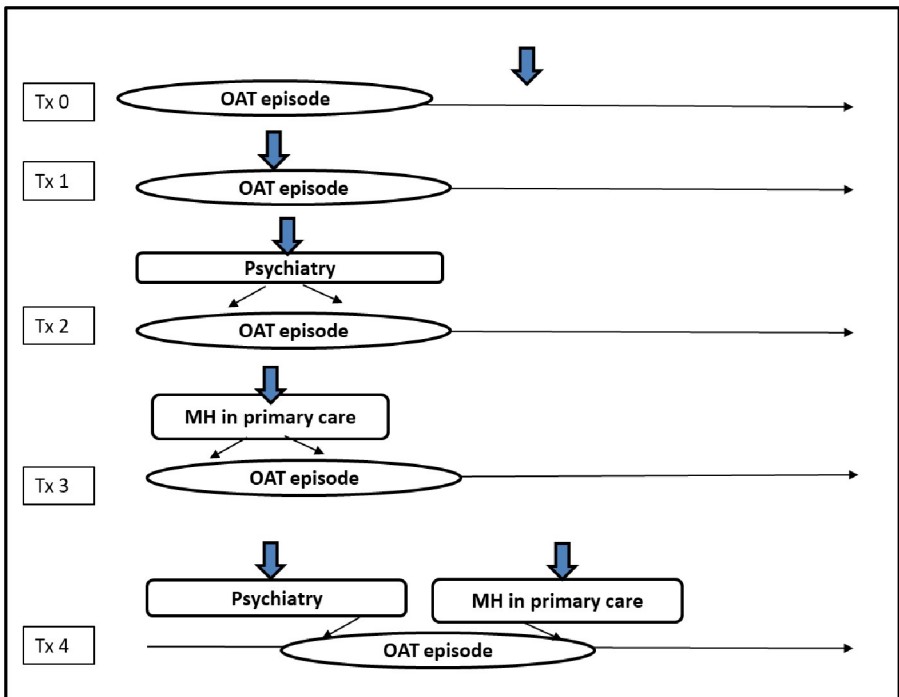

**Fig 3. Treatment groups (Tx).** $Tx_0$ = patients who were not actively engaged in OAT (control group). $Tx_1$ = patients who were actively engaged in OAT but had not received any additional mental health services from a physician. $Tx_2$ = patients who were actively engaged in OAT and had received mental health services from a psychiatrist during their active OAT episode of care. $Tx_3$ = patients who were actively engaged in OAT and had received mental health services from a primary care physician during their active OAT episode of care. $Tx_4$ = patients who were actively engaged in OAT and had received mental health services concurrently from a psychiatrist primary care physician during their active OAT episode of care.

Exposure to any of the mental health services groups (psychiatry, primary care, or both) was assumed for all individuals who had an out-patient mental health service encounter from one of the selected physicians identified by OHIP billing in any setting in the medical record. Billing codes are outlined in S2 Table.

Additionally, any individual assigned to the mental health services group would have a mental disorder-related International Classification of Disease (ICD)-9 or ICD-10 diagnostic codes and have received out-patient mental health service from a physician 30 days prior, during or 30 days after actively receiving OAT services. See S1 Table for the list of diagnoses and their related International Classification of Disease (ICD) codes.

## Baseline variables

The baseline variables were chosen based on previous literature, on the advice of our Patient and Family Advisory Group and on the data available in the databases used for this study. The baseline variables used included age, sex, location of residence, income quintile, human immunodeficiency virus status (HIV), deep tissue infections including endocarditis (OHIP fee code 429), osteomyelitis (OHIP fee code 730) and septic arthritis (OHIP fee code 711). Each variable was a baseline covariate. Therefore they were measured at the index date.

## Outcomes

Study outcomes were defined based on the need to assess the association of all-cause mortality, frequent ED visits, and hospitalizations as a function of concurrent disorders and mental health services and not as an exposure leading to an event. The outcomes chosen are important from a policy perspective [5]. All-cause mortality, frequent ED visits, and hospitalizations have been used as indicators of complexity in the OUD population in other studies [9, 17, 50, 51] and are regularly reported by the federal Government [5]. Additionally, both frequent ED visits and hospitalizations are metrics used by health system planners and funders in Ontario to understand gaps of services in communities [50–52].

## All-cause mortality

All-cause mortality was requested as a dichotomized variable and extracted from the RPDB database. The number of days to the death date from the index date for each patient in the cohort was calculated to determine whether a patient was deceased within the study period. If the patient had the event anytime between their index date and the end of the study period (December 31, 2016), a code of 1 (all-cause mortality) or 0 (no all-cause mortality) was assigned. The group with no all-cause mortality was used as the reference group for analysis.

## Frequent ED visits and hospitalizations

The NACRS database was used to identify ED visits. A patient was considered as having frequent ED visits if 1) contact with ED was after the index date, and 2) a patient had ten or more ED events in a publicly funded Ontario hospital within one year. Hospitalizations were extracted from the DAD database. Hospitalizations have been used in other studies and report an indicator of clinical complexity and vulnerability [53–55]. If a hospitalization discharge record appeared after a patient's index date in a publicly funded Ontario hospital, the hospitalization was counted.

## Statistical analysis

After checking for missing values, propensity scores were calculated using logistic regression analysis with age, sex, income quintile, the location of residence, deep tissue infections, and HIV status and treatment group and the outcome. Balancing covariates before further analyzing the data eliminates bias related to the treatment group assignment. Scores were then transformed into the inverse probability of treatment weights (IPTW) [56–58]. IPTW allows for the estimation of the average treatment. Adjusted weights were normalized [59]. Covariates were assessed (before and after weighting) using standardized differences (d) where d $\geq$ 10% indicates a clinically relevant difference.

Propensity score-weighted multivariate logistic regression models were used to determine the statistical significance of the associations between the treatment groups (Tx$_0$, Tx$_1$, Tx$_2$, Tx$_3$, Tx$_4$) and all-cause mortality. Parameter estimates were used to calculate odds ratios (OR) and their 95% confidence intervals (CI). Results were considered statistically significant, where p < 0.05.

We used the propensity matched data to run a negative binomial regression model to estimate the association between mental disorders, ED visits and hospitalizations. In a study comparing different models to evaluate health service utilization, the negative binomial regression model was found to be one of the most appropriate because it can address over dispersion that is typically observed in health care utilization data [60]. We used the regression coefficient to calculate OR and 95% CI [60, 61].

All statistical analysis was conducted on the remote server using SAS Version 9.4 [62]. Data was reviewed by ICES to insure privacy standards were met. This study is reported following the Strengthening the Reporting of Observational Studies in Epidemiology (STROBE) guidelines [63].

# Results

## Cohort characteristics

A total of 48,676 individuals in OAT between 2011 and 2016 and who had at least one concurrent mental disorder were included in the cohort. The most prevalent mental disorders identified in the cohort included: anxiety disorders, including obsessive compulsive disorder, and other anxiety-related disorders (60%), and mood disorders, including bipolar and other depressive-related disorders (20%). Of these patients with OUD and a diagnosed mental disorder, 1.6% (n = 771) were not actively engaged in OAT (Tx$_0$, the control group), 38.5% (n = 18,717) were actively engaged in OAT but did not access mental health services during their OAT episode of care (Tx$_1$), 8.1% (n = 3,939) received mental health services from a psychiatrist during their first year in OAT (Tx$_2$), 39.8% (n = 19,366) received mental health services from a primary care physician during their first year in OAT (Tx$_3$), and 12.1% (n = 5,883) received mental health services from both a psychiatrist and primary care physician during their first year in OAT (Tx$_4$). Approximately half of the patients were receiving mental health services during their first year in OAT, most from a primary care physician.

Before applying weights to the covariates, standardized differences indicated an imbalance between treatment groups for age, sex, place of residence, and deep tissue infections (Table 1). After propensity score weighting was applied to address the potential for treatment selection bias in the regression analyses, there were no significant differences across treatment groups, with all standardized differences (d) less than 10%. (Table 2). The 534 records with missing information on income quintile were re-assigned to the lowest income group, and those with missing information for location of residence were deleted (n = 3).

**Table 1. Unadjusted demographic and baseline measures.**

| Variables | Pre weight | | | | | | | | |
|---|---|---|---|---|---|---|---|---|---|
| | $Tx_0$ n = 771 (1.6) | d | $Tx_1$ n = 18,717 (38.5) | d | $Tx_2$ n = 3,939 (8.1) | d | $Tx_3$ n = 19,366 (39.8) | d | $Tx_4$ n = 5,883 (12.1) |
| **Sex N (%)** | | 24.79 | | 30.00 | | 13.78 | | 42.70 | |
| Male | 392 (50.8) | | 12,367 (40.4) | | 2,639 (67.0) | | 11,696 (60.4) | | 3,558 (39.5) |
| Female | 379 (49.2) | | 6,350 (35.2) | | 1300 (33.0) | | 7,670 (39.6) | | 2,325 (60.5) |
| **Age N (%)** | | 20.82 | | 4.97 | | 2.47 | | 1.62 | |
| 15–24 | 73 (9.5) | | 3,169 (16.9) | | 741 18.8) | | 3,639 (18.8) | | 1,104 (18.8) |
| 25–34 | 175 (22.7) | | 6,072 (32.4) | | 1,380 (35.0) | | 6,458 (35.4) | | 2,062 (35.1) |
| 35–44 | 147 (19.1) | | 4,134 (22.1) | | 861 (21.9) | | 4,271 (22.1) | | 1,299 (22.1) |
| 45–54 | 145 (18.8) | | 3,611 (19.3) | | 687 (17.4) | | 3,357 (17.3) | | 1,011 (17.2) |
| 55–64 | 98 (12.7) | | 1,343 (7.2) | | 231 (5.9) | | 1,334 (6.9) | | 341 (5.8) |
| 65+ | 133 (17.2) | | 388 (2.1) | | 38 (1.0) | | 307 (1.6) | | 66 (1.1) |
| **Geography N (%)** | | 11.13 | | 22.30 | | 13.58 | | 16.20 | |
| Northern Rural | 79 (10.3) | | 1,238 (6.6) | | 61 (1.6) | | 794 (4.1) | | 68 (1.2) |
| Northern Urban | 63 (8.2) | | 2,442 (13.1) | | 191 (4.9) | | 1,621 (8.4) | | 216 (6.7) |
| Southern Rural | 86 (11.2) | | 1,491 (8.0) | | 302 (7.7) | | 1,775 (9.2) | | 362 (6.2) |
| Southern Urban | 543 (70.4) | | 13,546 (72.4) | | 3,385 (85.9) | | 15,176 (78.4) | | 5,237 (89.0) |
| **Income Quintile N (%)** | | 3.42 | | 5.38 | | 6.28 | | 1.44 | |
| 1 (lowest) | 260 (33.7) | | 6,241 (33.3) | | 1,521 (38.6) | | 6,509 (33.6) | | 2,023 (34.4) |
| 2 | 176 (22.8) | | 4,322 (23.1) | | 894 (22.7) | | 4,185 (21.6) | | 1,309 (22.3) |
| 3 | 139 (18.0) | | 3,363 (18.0) | | 682 (17.3) | | 3,497 (18.1) | | 1,009 (17.2) |
| 4 | 91 (11.8) | | 2,710 (14.5) | | 468 (11.9) | | 2,888 (14.9) | | 857 (14.6) |
| 5 | 105 (13.6)) | | 2,081 (11.1) | | 374 (9.5) | | 2,287 (11.8) | | 685 (11.6) |
| **HIV** | <10 | 2.80 | 117 (0.6) | 7.20 | 50 (1.3) | 4.90 | 162 (0.8) | 2.10 | 56 (1.0) |
| **Deep Tissue infections** | 58 (7.5) | 21.40 | 530 (2.8) | 2.30 | 125 (3.2) | 0.00 | 612 (3.2) | 6.30 | 259 (4.4) |

## Outcomes

**All-cause mortality.** The highest percentage of patients with all-cause mortality was among those not actively engaged in an OAT episode of care ($Tx_0$ = 11.9%), compared to patients actively engaged in OAT who had not received mental health services ($Tx_1$ = 4.6%), patients who received mental health services from a psychiatrist ($Tx_2$ = 5.6%), patients who received mental health services from a primary care physician ($Tx_3$ = 4.9%), and patients who received mental health services from both a psychiatrist and a primary care physician ($Tx_4$ = 6.0%). Active engagement in OAT ($Tx_1$) was associated with decreased likelihood of all-cause mortality (OR = 0.4, 95% CI 0.3–0.4) compared to no active engagement in OAT. Similarly, mental health services from a psychiatrist ($Tx_2$) and mental health services from a primary care physician ($Tx_3$) during the first year of OAT was associated with decreased likelihood of all-cause mortality (OR = 0.4, 95% CI 0.4–0.6; OR = 0.4, 95% CI 0.3–0.5). As well OAT, mental health services from a psychiatrist and a primary care physician concurrently while in OAT ($Tx_4$) was associated with decreased likelihood of all-cause mortality (OR = 0.5, 95% CI 0.5–0.6). See Fig 4 and S3 Table for detailed results.

**ED visits.** For patients in the treatment groups, one, two and three (patients engaged in OAT with no mental health support (Tx1), (patietns engaged in OAT and receiving services from a psychiatrist ($Tx_2$) and (patietns engaged in OAT and mental health services from a primary care physician (Tx3)) the estimated rate of ED visits per year was lower than in the control group (patients not engaged in OAT) (Tx1, OR = 0.1, 95% CI 0.0–0.2; Tx2, OR = 0.9, 95% CI 0.8–0.9), Tx3, OR = 0.7, 95% CI 0.7–0.8). However, in treatment group four (patients

**Table 2. Propensity score weighted demographic and baseline measures.**

| Variables | Post weight | | | | | | | | |
|---|---|---|---|---|---|---|---|---|---|
| | Tx$_0$ n = 759.5 (1.7) | d | Tx$_1$ n = 18,609.7 (38.4) | d | Tx$_2$ n = 3,912.2 (8.1) | d | Tx$_3$ n = 19,266.1 (39.8) | d | Tx$_4$ n = 5,867.8 (12.1) |
| **Sex N (%)** | | 4.49 | | 1.00 | | 1.20 | | 1.22 | |
| Male | 462.8 (60.9) | | 11,748.8 (63.1) | | 2,486.7 (63.6) | | 12,138.4 (63.0) | | 3,730.1 (63.6) |
| Female | 296.7 (39.1) | | 6,860.9 (36.9) | | 1,425.4 (36.4) | | 7,127.7 (37.0) | | 2,137.7 (36.4) |
| **Age N (%)** | | 0.58 | | 0.44 | | 0.38 | | 0.47 | |
| 15–24 | 138.4 (18.2) | | 3,321.7 (17.9) | | 699.0 (17.9) | | 3,449.5 (17.9) | | 1,022.8 (17.4) |
| 25–34 | 2,48.7 (32.8) | | 6,146.8 (33.0) | | 1,300.4 (33.2) | | 6,381.0 (33.1) | | 1,963.9 (33.5) |
| 35–44 | 169.9 (22.4) | | 4,107.0 (22.1) | | 857.7 (22.0) | | 4,242.4 (22.0) | | 1,283.1 (21.9) |
| 45–54 | 137.5 (18.1) | | 3,377.4 (18.1) | | 706.3 (18.1) | | 3,491.6 (18.1) | | 1,075.8 (18.3) |
| 55–64 | 49.6 (6.5) | | 1,291.5 (6.9) | | 276.2 (7.1) | | 1,332.7 (7.0) | | 411.0 (7.0) |
| 65+ | 15.4 (2.0) | | 365.2 (2.0) | | 72.5 (1.7) | | 368.9 (1.9) | | 111.1 (1.9) |
| **Geography N (%)** | | 1.75 | | 0.60 | | 0.70 | | 0.43 | |
| Northern Rural | 37.9 (5.0) | | 860.8 (4.6) | | 181.5 (4.6) | | 888.5 (4.6) | | 280.5 (4.8) |
| Northern Urban | 65.6 (8.6) | | 1,738.1 (9.3) | | 348.2 (8.9) | | 1,795.8 (9.3) | | 552.5 (9.4) |
| Southern Rural | 60.3 (8.0) | | 1,540.6 (8.3) | | 323.6 (8.3) | | 1,588.2 (8.2) | | 481.3 (8.2) |
| Southern Urban | 595.6 (78.4) | | 1,4470.2 (77.8) | | 3,058.8 (78.2) | | 14,993.6 (77.8) | | 4,553.5 (77.6) |
| **Income Quintile N (%)** | | 3.54 | | 0.40 | | 0.36 | | 0.63 | |
| 1 (lowest) | 279.6 (36.8) | | 6,339.0 (34.1) | | 1,324.2 (33.9) | | 6,554.7 (34.0) | | 1,957.6 (33.4) |
| 2 | 176.6 (23.3) | | 4,167.5 (22.4) | | 880.3 (22.5) | | 4,308.6 (22.4) | | 1,329.7 (22.7) |
| 3 | 117.7 (15.5) | | 3,312.8 (17.8) | | 708.9 (18.1) | | 3,434.4 (17.8) | | 1,050.4 (17.9) |
| 4 | 108.5 (14.3) | | 2,675.7 (14.4) | | 555.5 (14.2) | | 2,777 (14.4) | | 843.1 (14.4) |
| 5 (highest) | 77.0 (10.3) | | 2,114.7 (11.4) | | 443.3 (11.3) | | 2,190.8 (11.4) | | 687 (11.7) |
| **HIV** | 8.0 (1.1) | 3.10 | 11.4 (0.8) | 0.00 | 32.2 (0.8) | 0.00 | 153.9 (0.8) | 0.00 | 48.8 (0.8) |
| **Deep Tissue infections** | 40.5 (5.3) | 9.90 | 605.2 (3.3) | 0.60 | 132.2 (3.4) | 0.60 | 628.7 (3.3) | 0.60 | 188.3 (3.2) |

Tx$_0$ = no OAT

Tx1 = OAT only

Tx2 = OAT and psychiatry

Tx3 = OAT and mental health services from a primary care provider

Tx3 = OAT, psychiatry and mental health services from a primary care provider

engaged in OAT who received mental health services from a psychiatrist and primary care physician at the same time) the estimated rate of ED visits per year was higher than in the control group (Tx4, OR = 1.3, 95% CI 1.2–1.4).

**Hospitalizations.** For patients in all the treatment groups (Tx$_1$, Tx$_2$, Tx$_3$, Tx$_4$) the estimated rate of hospitalizations per year was lower than in the control group (Tx$_1$, OR = 0.6, 95% CI 0.1–0.7; Tx$_2$, OR = 0.4, 95% CI 0.2–0.9), Tx$_3$, OR = 0.8, 95% CI 0.7–0.9; Tx$_4$, OR = 0.5, 95% CI 0.4–0.6). See S3 Table for detailed results.

## Discussion

Similar to findings in other studies, we found that OAT alone was associated with positive outcomes for patients with OUD [64–67]. In this study, we grouped patients based on their access to physician-based mental health services during their first year of OAT. We found that receiving mental health services from a psychiatrist, a primary care physician, or both within the first year of OAT was associated with a decreased likelihood of death from any cause. However,

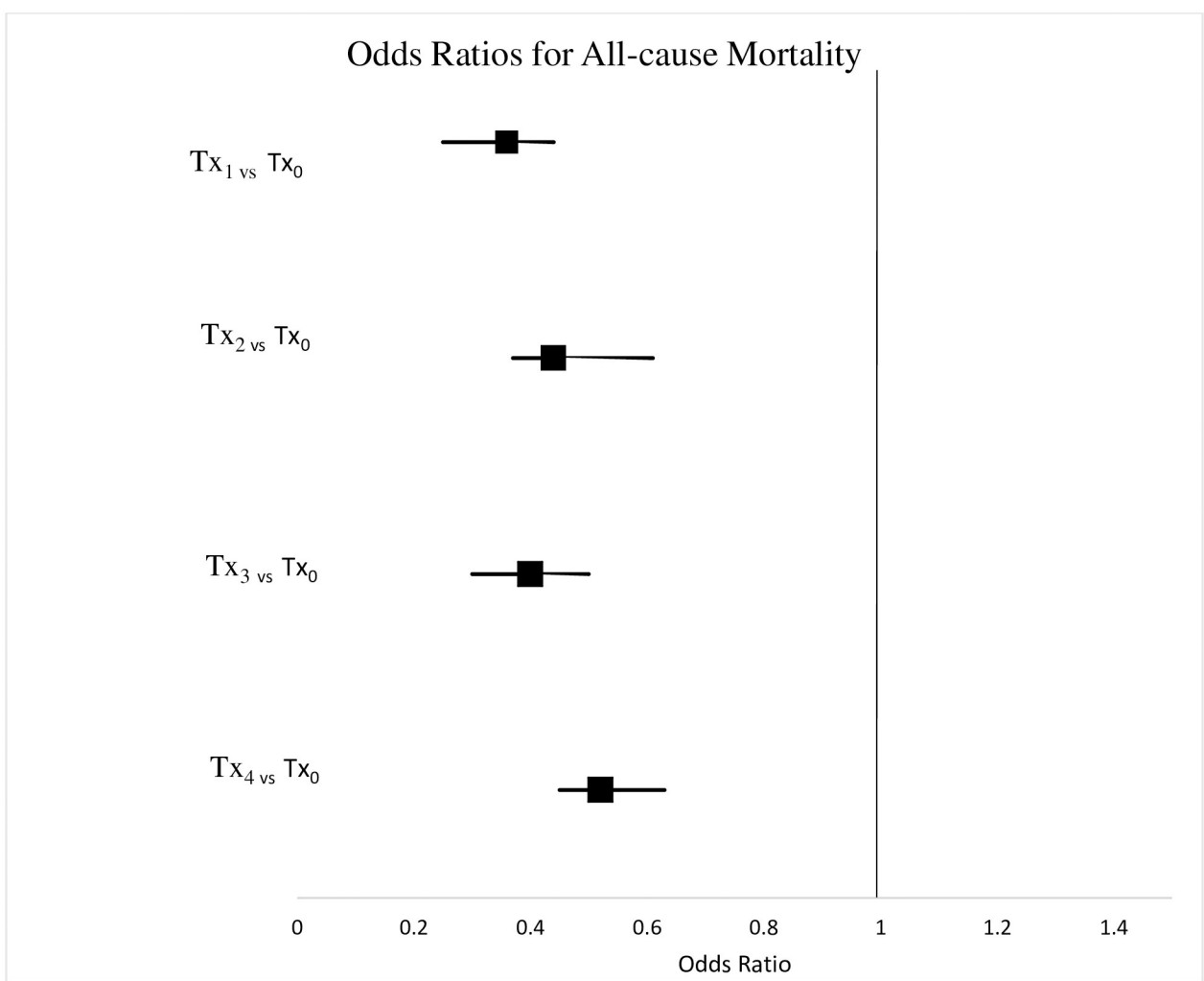

**Fig 4. Odds ratio for all-cause mortality by treatment group.**

we also found that those accessing mental health services during the first year of OAT was associated with frequent acute care use. Additionally, we observed imbalanced covariates between treatment groups with regards to gender, age, income, geography, and infectious disease status.

## Cohort characteristics

We observed imbalanced covariates between treatment groups for sex, age, income, geography, and infectious disease status. The imbalance observed between the five treatment groups indicated that patient characteristics influenced the treatment group assignment for this study. Importantly, some of the most significant differences between groups were observed between the geographic regions. For instance, northern rural areas had a lower proportion of patients accessing concurrent mental health services during their first year of OAT. Such results may be explained by the lack of availability and limited range of services in northern regions, as well as the travel distance. Before creating the regression model to evaluate the relationship

between treatment groups and outcomes, we created propensity scores and weighted variables to address the treatment selection bias identified above.

## All-cause mortality

All-cause mortality has been used as an indicator of the severity of the opioid crisis across Ontario in both rural and urban areas. Opioid poisoning- related deaths have more than tripled between 1999 and 2016 [68]. In a recent cohort study examining the causes of death among methadone patients in Scotland, 59% experienced non-drug related deaths [69]. Many mortality related studies focus on opioid-related deaths. However, our use of all-cause mortality allowed us to explore mortality more broadly. By measuring all-cause mortality, we were able to represent the cumulative influences of an unstable lifestyle, such as engagement in risk behaviours, poly-substance use, difficulties with relationships, impulsivity, and difficulties with access to care of individuals with OUD [17]. We found that all treatment groups were associated with reduced odds of all-cause mortality when compared against the control group (patients not actively engaged in OAT). Our finding is consistent with the previous literature [17, 70, 71]. For instance, in a recent meta-analysis based on 16 studies, Ma et al., found that untreated participants had higher all-cause mortality (RR = 2.56, 95% CI, 1.72–3.80) and overdose mortality (RR = 8.10, 95% CI, 4.48–14.66]) when compared to those actively engaged in OAT [71].

## Elevated use of ED and hospitals

In addition to ACM, ED visits and hospitalizations are reported regularly by the government of Canada to measure the severity of the opioid crisis [5, 9]. In 2016, a report by the Ontario Drug Policy Research Network (ODPRN) demonstrated that opioid-related hospital admissions in the province vary from 0.5 admissions per 10,000 residents in the southern region and up to 2.4 admissions per 10,000 residents in northern regions. Similarly, rates of opioid-related frequent ED visits varied nearly 4-fold from 1.2 visits per 10,000 residents in southern regions to 4.5 visits per 10,000 residents in the north [9]. In our study, mental health services from a psychiatrist and a family physicians concurrently during OAT was associated with frequent frequent ED visits. While potentially counter-intuitive, this finding may be explained by the fact that more complex patients need more services. High utilization may stem from limited access to sufficient mental health services in the community [10, 17, 18, 72], forcing patients to access the health services they need in the ED. For many patients with OUD, despite being enrolled in OAT, the ED may be the primary route to accessing treatment for mental disorders. Studies have shown that mental health services, including case management, could decrease ED use [52, 73, 74]. However, in rural areas, the ED alone might be the only point of contact available for patients with mental disorders and OUD. Therefore, it is important to highlight that, although the ED is not the ideal type of service for patients with chronic health issues, it is an important resource to ensure that patients are getting access to some type of health care in areas where resources are limited.

Other authors have demonstrated that the most effective ways to improve patient outcomes are intermediate levels of OAT [75] and mental health services offered on-site within an OAT program [66, 76] within the first six months of an OAT episode of care [77]. Although our results indicate that complex patients are accessing mental health services from physicians, we were not able to identify whether their access to mental health services was coordinated in a manner that was conducive to their health and social needs. Importantly, our results suggest that concurrent or coordinated care may not be needed for all patients, but may have a

significant impact on some of the most complex patients seeking treatment for OUD. More research is required to explore this concept further.

## Strengths and limitations

Our study has notable strengths and limitations. The use of a large database allowed us to robustly examine the issue of OUD and concurrent mental health services at a population level. Our use of a population health approach enabled us to broadly analyze and compare a specified group of patients across Ontario as well as to conduct sophisticated analysis that replicates a randomized clinical trial approach using retrospective administrative data. This type of analysis would be very difficult to achieve if we studied a more targeted group in a smaller scale study. However, using secondary data has its limitations. The use of physician billing is dependent on the accuracy and reliability of recording practices.

Moreover, factors such as service volume, quality of care, location, and coordination between physicians and organizations are missed in this type of population health approach. Additionally, we were limited only to examine OHIP billed mental health services, which excluded the exploration of private, community and Federally-funded health services—such as mental health services provided in Indigenous communities—as well as any mental health services funded a provincial ministry other than the Ministry of Health and Long Term Care —such as in correctional facilities. Lastly, it is important to consider that, for those patients where mortality occurred during the first year of OAT, the likelihood of frequent ED use or hospital admission is reduced. However, the function of time is a modest bias since we observed an increase in frequent ED visits and hospitalizations in the groups with the highest mortality. It is also important to consider that if any crucial variables have been omitted to a propensity model, the groups may remain unbalanced, and the results of the study can be biased [78, 79]. Therefore, results significant results must be interpreted critically within the context of the population of interest to determine implications for clinical services or health system design and evaluation.

## Conclusion

The findings of our study are consistent with the literature indicating that OAT is an effective intervention available to improve clinical and health system outcomes for patients with OUD [22, 65, 80–83], even those with concurrent disorders. Also, our findings contribute to the literature, which supports the view that OAT and concurrent mental health services are consistently associated with improved clinical outcomes, more specifically, all-cause mortality. Our study does not suggest that coordinated care should be implemented for all OAT patients. However, it should be considered for those complex patients who are high users of health care services. Results may be generalizable in regions where OAT programs and health care regulations are similar to those in Ontario. Knowing the high prevalence of mental disorders in the OUD population, physicians and policymakers must understand the potential importance of access to mental health services.

## Supporting information

**S1 Table. Definition and ICD 9 and ICD10 codes for mental health conditions.**
(DOCX)

**S2 Table. Mental health service OHIP fee codes.**
(DOCX)

**S3 Table. Study outcomes related to concurrent physician-base mental health services and OAT.**
(DOCX)

## Acknowledgments

We thank IC/ES Data Analytic Services for assistance with data extraction and database set up. We also thank the Patient and Family Advisory Committee members for sharing their stories and helping to guide the research project.

## Author Contributions

**Conceptualization:** Kristen A. Morin, Joseph K. Eibl, Joseph M. Caswell, Brian Rush, Christopher Mushquash, Nancy E. Lightfoot, David C. Marsh.

**Formal analysis:** Kristen A. Morin, Joseph M. Caswell.

**Funding acquisition:** David C. Marsh.

**Investigation:** Kristen A. Morin, David C. Marsh.

**Methodology:** Kristen A. Morin, Joseph K. Eibl, Joseph M. Caswell, Brian Rush, Nancy E. Lightfoot.

**Project administration:** Kristen A. Morin.

**Supervision:** Joseph K. Eibl, Brian Rush, Christopher Mushquash, Nancy E. Lightfoot, David C. Marsh.

**Validation:** Joseph M. Caswell.

**Writing – original draft:** Kristen A. Morin.

**Writing – review & editing:** Kristen A. Morin, Joseph K. Eibl, Brian Rush, Christopher Mushquash, Nancy E. Lightfoot, David C. Marsh.

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
