## [Decision Letter · Decision Letter 0]

6 Oct 2020

PONE-D-20-25138

Evaluating the Effectiveness of Concurrent Opioid Agonist Treatment and Physician-Based Mental Health Services for Patients with Mental Disorders in Ontario, Canada

PLOS ONE

Dear Dr. Marsh,

Thank you for submitting your manuscript to PLOS ONE. After careful consideration, we feel that it has merit but does not fully meet PLOS ONE’s publication criteria as it currently stands. Therefore, we invite you to submit a revised version of the manuscript that addresses the points raised during the review process.

We look forward to receiving your revised manuscript.

Kind regards,

Sungwoo Lim, DrPH

Academic Editor

PLOS ONE

Journal Requirements:

2. Please include your tables as part of your main manuscript and remove the individual files. Please note that supplementary tables (should remain/ be uploaded) as separate "supporting information" files

3.Thank you for stating the following in the Acknowledgments Section of your manuscript:

[We thank our funders (Northern Ontario Academic Medical Association) through the Academic

394 Funding Plan Innovation Fund for their support in completing this project, grant # A-17-05.]

 [n/a]

4.Thank you for stating the following in the Competing Interests section:

[Dr. David Marsh maintains the following roles: Chief Medical Director at CATC (Canadian Addiction Treatment Center), opioid agonist therapy provider. Dr. Marsh has no ownership stake in the CATC as a stipendiary employee. We do not foresee any conflict of interest as data will be made freely available to the public and neither the CATC, nor the Universities, have the ability to prevent publication and dissemination of knowledge. The other authors have no conflicts declared. ].

5. Please ensure that you refer to Figure 4 in your text as, if accepted, production will need this reference to link the reader to the figure.

6. Please upload a copy of Figure 5, to which you refer in your text on page 20. If the figure is no longer to be included as part of the submission please remove all reference to it within the text.

Reviewers' comments:

Reviewer's Responses to Questions

**Comments to the Author**

1. Is the manuscript technically sound, and do the data support the conclusions?

Reviewer #1: Partly

Reviewer #2: Yes

2. Has the statistical analysis been performed appropriately and rigorously? 

Reviewer #1: Yes

Reviewer #2: Yes

3. Have the authors made all data underlying the findings in their manuscript fully available?

Reviewer #1: No

Reviewer #2: Yes

4. Is the manuscript presented in an intelligible fashion and written in standard English?

Reviewer #1: Yes

Reviewer #2: Yes

5. Review Comments to the Author

Reviewer #1: Thanks for allowing me to review this interesting manuscript, which aims to advance the OUD/concurrent disorder field.

There are many strengths to this study, including the importance of the topic, the organization of the manuscript, the methodological approaches, and the use of flow diagrams to help structure the flow of data. In my opinion, the authors have done an excellent job in outlining the context for such a study as well as in leveraging large linked datasets to conduct an interesting study. The analytic component of the study, particularly the description of the methods, was extremely well done.

However, there are some serious limitations that blunt my enthusiasm for this paper, and prevent my recommending it for further consideration.

1) Novelty. The conclusions of the manuscript largely agree with consensus literature, which the authors highlight throughout the text. Ultimately, the conclusion appears to be that concurrent disorder treatment improves outcomes for people with OUD? If I am incorrect on this, my apologies. But to that end, I am not sure what it is adding because research must add something to the pool to justify publication, etc.

2) Heterogeneity. One of the strengths oft he study is the large sample size. With the advent of large, administrative datasets, like those of IC/ES, it is possible to conduct gigantic observational studies, and one can use propensity score methods to adjust, quite well. However, an issue remains related to heterogeneity. Despite how close the authors are able to approximate the two populations, I question whether this is a realistic approach given the substantial heterogeneity in the OUD population. As well, I am not sure how useful it is to psychiatrists, clinicians, or addiction medicine specialists if there is limited emphasis on "what" concurrent disorder treatment is helpful, and for "which" concurrent disorders.

3) Secondary data use. This relates to the above point, which stems on the use of secondary data. IC/ES uses a number of proxy measures to estimate exposures and outcomes from administrative datasets. This is akin to handing receipts to an accountant and asking them to determine the financial narrative of someone's life. There are advantages, and disadvantages to this approach, as highlighted by the authors. The main disadvantage the authors encounter is a lack of context, as the study provides a very narrow view of a very complex issue. In order to ensure that data fits a model and to agree with statistical assumptions, many outliers and other aspects of the data have to be trimmed and curtailed. To that end, we see in the flow diagrams that a large sample is significantly cut down by the end. While the authors discuss the limitations, highlighting them is not enough. To that end, there are some serious methodological issues with several of the variables used. For example, using OHIP billing as a proxy for a diagnosis or for providing a treatment for a concurrent disorder is very problematic, as the billing system is raked with discrepancies between what is billed for and what is done. This will never be captured in a study because studies will only cite accuracy of the system; however, this is not a realistic measure of the quality of the data. With that said, I am not sure what a superior method for this would look like that would provide as large of a sample size. Ultimately, many of these issues are generalized limitations of using large administrative data sets to answer clinical questions and unfortunately, the authors have not providing a convincing rationale that there approach is justified given these limitations.

4) Generalizability. My other concern is that the study appears to exclude the highest risk people with severe OUD and severe mental health disorders, which are not "in the system" so to speak. Those who do not have OHIP/Ontario ID, who are marginally housed, who live on reserves, who live in rural and remote regions of Ontario, are left out of the administrative dataset access. Most clinician would argue that the population excluded by this study is the precise population for whom the least is known. Thus, I wish that there was a way to capture those who are understudied, as that would certainly be a novel area for researchers to explore, particularly with the types of methods the authors present. I think this points to a serious issue with selection bias, which limits the generalizability of the authors results; this is particularly worrisome given that the study was already limited in what it could contribute to the literature given that this topic is fairly well-studied. As well, research that presents a more realistic, "real-world" picture of a clinical scenario is more valuable, so, the pertinent exclusions from this study make me wonder if the population that this study explored even "exists" in everyday clinical practice. With this in mind, concurrent substance use disorder is a key comorbidity (e.g., in addition to OUD, stimulant, benzodiazepine, alcohol, etc.), and I am not sure if the authors have appropriately considered this aspect in the analyses. I think this would have a major impact on not only response to OAT, but also, the response to concurrent mental health treatment (as well as the approach to treatment).

5) I do not necessarily disagree with the findings, but, I think re-packaging a fairly well-studied topic justifies a higher expectation for methodological rigour and requires researchers to develop innovative strategies to approach the limitations.

Reviewer #2: The manuscript entitled, “Evaluating the Effectiveness of Concurrent Opioid Agonist Treatment and Physician-Based Mental Health Services for Patients with Mental Disorders in Ontario, Canada,” presents findings of a large-scale administrative data study evaluating the impact of co-occurring mental health disorders and OUD. Strengths of this paper include evaluation of a scientifically and clinically important question a robust analytic plan. Overall weaknesses include lack of context and rationale for specific aspects of the study (e.g., physician-only mental health care) and lack of discussion/evaluation of the heterogeneous landscape mental health disorders and their interaction with OUD as it relates to the outcomes. These weaknesses make the manuscript in its current form difficult to place in context of other work on co-occurring OUD and mental health disorders. See below for comments related to specific issues.

1) More discussion of the types of commonly co-occurring mental disorders, and the types of complexities that can result (e.g., medical symptom exacerbation, treatment challenges, etc) would strengthen the introduction and clarify the hypotheses (lines 63-66).

2) Rationale for only including mental health care provided by psychiatrists and/or family physicians is needed. Why is physician-based mental health care examined but not other types?

3) Rationale for the covariates also needs to be provided. What is the scientific rational for including these medical conditions, but not others?

4) Results: the types of co-occurring mental disorders should be described.

5) Line 287: Please cite the work referred to in the statement, “Similar to findings in other studies…”

6. PLOS authors have the option to publish the peer review history of their article (what does this mean?). If published, this will include your full peer review and any attached files.

Reviewer #1: No

Reviewer #2: **Yes: **Anthony Ecker

---

## [Author Response · Author response to Decision Letter 0]

9 Oct 2020

As requested by the editorial team, we have included a rebuttle letter that responds to each point raised by the academic

editor and reviewers labelled 'Response to Reviewers'. Due to formatting, I was not able to copy/ paste the letter here.

---

## [Decision Letter · Decision Letter 1]

30 Oct 2020

PONE-D-20-25138R1

Evaluating the Effectiveness of Concurrent Opioid Agonist Treatment and Physician-Based Mental Health Services for Patients with Mental Disorders in Ontario, Canada

PLOS ONE

Dear Dr. Marsh,

Thank you for submitting your manuscript to PLOS ONE. After careful consideration, we feel that it has merit but does not fully meet PLOS ONE’s publication criteria as it currently stands. Therefore, we invite you to submit a revised version of the manuscript that addresses the points raised during the review process.

We look forward to receiving your revised manuscript.

Kind regards,

Sungwoo Lim, DrPH

Academic Editor

PLOS ONE

Reviewers' comments:

Reviewer's Responses to Questions

**Comments to the Author**

1. If the authors have adequately addressed your comments raised in a previous round of review and you feel that this manuscript is now acceptable for publication, you may indicate that here to bypass the “Comments to the Author” section, enter your conflict of interest statement in the “Confidential to Editor” section, and submit your "Accept" recommendation.

Reviewer #2: All comments have been addressed

2. Is the manuscript technically sound, and do the data support the conclusions?

Reviewer #2: Yes

3. Has the statistical analysis been performed appropriately and rigorously? 

Reviewer #2: Yes

4. Have the authors made all data underlying the findings in their manuscript fully available?

Reviewer #2: Yes

5. Is the manuscript presented in an intelligible fashion and written in standard English?

Reviewer #2: Yes

6. Review Comments to the Author

Reviewer #2: The revisions have resulted in a stronger manuscript. Please see below for remaining issues.

Line 63-66. Although authors do describe psychosocial and health problems in the next paragraph, information on the types of mental disorders referred to remains missing (e.g., depression, anxiety, psychotic disorders, etc)

Line 146, the use of 1st person is out of place.

7. PLOS authors have the option to publish the peer review history of their article (what does this mean?). If published, this will include your full peer review and any attached files.

Reviewer #2: **Yes: **Anthony H Ecker PhD

---

## [Author Response · Author response to Decision Letter 1]

10 Nov 2020

Dear reviewers and editorial team,

Thank you for your thoughtful review. We sincerely believe that the revisions have resulted in a stronger manuscript. Please see our response and the reference to the changes in the manuscript summarized the 'Response to Reviewers' document attached.

Kindest regards,

Dr. David C. Marsh

---

## [Editor Report · Decision Letter 2]

19 Nov 2020

Evaluating the Effectiveness of Concurrent Opioid Agonist Treatment and Physician-Based Mental Health Services for Patients with Mental Disorders in Ontario, Canada

PONE-D-20-25138R2

Dear Dr. Marsh,

We’re pleased to inform you that your manuscript has been judged scientifically suitable for publication and will be formally accepted for publication once it meets all outstanding technical requirements.

Kind regards,

Sungwoo Lim, DrPH

Academic Editor

PLOS ONE
---

## [Editor Report · Acceptance letter]

2 Dec 2020

PONE-D-20-25138R2 

Evaluating the Effectiveness of Concurrent Opioid Agonist Treatment and Physician-Based Mental Health Services for Patients with Mental Disorders in Ontario, Canada 

Dear Dr. Marsh:

I'm pleased to inform you that your manuscript has been deemed suitable for publication in PLOS ONE. Congratulations! Your manuscript is now with our production department. 

Kind regards, 

on behalf of

Dr. Sungwoo Lim 

Academic Editor

PLOS ONE